# The Impact of Al*x*Ga1−*x*N Back Barrier in AlGaN/GaN High Electron Mobility Transistors (HEMTs) on Si*x*-Inch MCZ Si Substrate

**H.Y. Wang [1], H.C. Chiu [1,*], W.C. Hsu [2], C.M. Liu [2], C.Y. Chuang [2], J.Z. Liu [2] and Y.L. Huang [2]**

[1]   Department of Electronics Engineering, Chang Gung University, Taoyuan City 33302, Taiwan; jacky05162002@hotmail.com.tw

[2]   Innovation Technology Research Center, GlobalWafers Co., Ltd., Hsinchu City 300091, Taiwan; CHUCK@sas-globalwafers.com (W.C.H.); calvin@sas-globalwafers.com (C.M.L.); jeromy@sas-globalwafers.com (C.Y.C.); jack.liu@sas-globalwafers.com (J.Z.L.); Allen.Huang@sas-globalwafers.com (Y.L.H.)

[*]   Correspondence: hcchiu@mail.cgu.edu; Tel.: +886-3-2118800-3350

**Abstract:** In this study, AlGaN/GaN high electron mobility transistors (HEMTs) with AlGaN back barriers (B.B.) were comprehensively investigated based on the different Al mole fractions and thicknesses in the design of the experiments. It was shown that the off-state leakage current can be suppressed following an increase of the Al mole fraction due to the enhancement of the back barrier height. Increasing the AlGaN thickness deteriorated device performance because of the generation of lattice mismatch induced surface defects. The dynamic on-resistance ($R_{ON}$) measurements indicated that the Al mole fraction and thickness of the B.B. both affected the buffer trapping phenomenon. In addition, the thickness of B.B. also influenced the substrate heat dissipation ability which is also a key index for high power RF device applications.

**Keywords:** GaN; HEMT; microwave device; AlGaN back barrier

## 1. Introduction

Due to the numerous advancements of the IMT-2020 standard, the global competition to launch fifth-generation (5G) sub-six and mmWave frequencies could provide a distinct base-station market opportunity for the gallium nitride (GaN) high electron mobility transistor (HEMT) as an alternative to the silicon laterally diffused metal oxide semiconductor (LDMOS) [1–5]. For high efficiency and high linearity consideration, conventional Si CMOS processes are not suitable for emerging 5G applications due to their low operation drain voltage, where GaN HEMT can be operated under 50V. Traditionally, the GaN heterostructure devices have been grown on semi-insulating silicon carbide (SiC) due to its excellent thermal conductivity and low RF signal loss [6–8]. However, semi-insulating SiC substrate is still expensive but produced in high volume. A different approach to obtain a GaN RF HEMT is the adoption of 6-inch or 8-inch high resistivity float zone (FZ) Si substrate. The performance and reliability benefits of GaN-on-Si HEMTs for RF applications have been well demonstrated since the 2006 qualification of GaN-on-Si technology and this technology was implemented in 4G LTE and the 5G wireless base-station infrastructure successfully [4]. However, the cost and the crystal formation duration of high resistivity FZ silicon obtained by the vertical zone melting method are still two disadvantages for industrial applications.

In addition, the substrate mechanical strength of FZ Si is also lower than conventional Czochralski (CZ) Si thus the substrate bowing phenomena of GaN heterostructure on FZ Si also limits its potential for high volume production [9–11]. Recently, high resistivity magnetic Czochralski (MCz) silicon (>2000 $\Omega \cdot$cm) has been found to be of high interest for the development of radio-frequency RFCMOS and SOI technologies, since this material significantly suppresses the RF noise coupling from the substrate and transmission line intermodulation parasitic, together with a high substrate mechanical strength performance. In this work, we comprehensively studied the GaN RF HEMT on high resistivity MCZ with the AlGaN back barrier (B.B.) optimization. The usage of a B.B. structure is an alternative solution for reducing the short-channel effects and improving the carrier confinement in two- dimensional electron gas (2DEG) for AlGaN/GaN HEMT [12–16]. In addition, the buffer related electron trapping effect can also be minimized. Moreover, AlGaN back barrier layer also forms a thermal barrier layer which influences the device linearity and thermal induced current reduction at high power operation. The Al mole fraction and the thickness of the AlGaN back layer were two key parameters studied and investigated in this work.

## 2. Device Fabrication

The AlGaN/GaN RF HEMT with AlGaN B.B. structures were grown using the metal organic chemical vapor deposition (MOCVD) method on MCz-Si substrate. After an AlN nucleation layer, a 1.5 µm iron (Fe) doping GaN buffer layer was grown followed by the AlGaN back barrier. To study the impact of the AlGaN B.B., the design of experiments (DOE) of AlGaN B.B. were of different Al mole fractions (3%, 5%, and 8%) and thicknesses (40, 50, and 60 nm). As to the active layers of these samples, a 1-nm-thick AlN layer was grown in all the samples on top of the 300 nm GaN channel, followed by a 18 nm-thick $Al_{0.24}Ga_{0.76}N$ barrier layer and a 2 nm GaN cap layer. The thin AlN interlayer between the $Al_{0.24}Ga_{0.76}N$ barrier layer and the GaN channel contributes to reduce interface roughness and enhance the carrier mobility in 2-DEG. The Hall measurement result of these back barrier conditions was quite similar. The mobility range of devices were about 1628 to 1756 $cm^2$/V·s and the concentration of devices was about $1.14 \times 10^{13}$/$cm^2$ to $1.37 \times 10^{13}$/$cm^2$.

Figure 1a illustrates the cross-section of the fabricated device and the proposed epitaxial structures. Figure 1b illustrates the one-dimensional band diagrams of Schrödinger–Poisson simulation for the devices with various AlGaN B.B. designs. The higher potential of the $Al_xGa_{1-x}N$ back barrier is expected to improve the 2DEG carrier confinement [10]. As to the device fabrication, the active region was protected by a photoresist, and the mesa isolation region was removed in a reactive ion etching chamber using $BCl_3 + Cl_2$ mixed-gas plasma. Ohmic contacts were prepared using electron beam evaporation involving a multilayered Ti/Al/Ni/Au (30 nm/125 nm/50 nm/200 nm) sequence, which was followed by rapid thermal annealing at 850 °C for 30 s in a nitrogen-rich environment. After ohmic formation, the Ni/Au (15/330 nm) gate metal was evaporated.

Although, the AlGaN back barrier can improve the carrier confinement in two-dimensional electron gas (2DEG), it also causes a lattice mismatch phenomenon between channel and buffer. To investigate the dislocation with various B.B. layer designs, the structure lattice dislocation was calculated by the XRD FWHM results as following [17,18]:

$$N_{\text{screw}} \quad \frac{\text{FWHM}_{002}^2}{4.35 \times b_{\text{screw}}^2} \quad N_{\text{edge}} \quad \frac{\text{FWHM}_{102}^2}{4.35 \times b_{\text{edge}}^2} \tag{1}$$

$$N_{\text{total}} = N_{\text{screw}} + N_{\text{edge}} \tag{2}$$

where $N_{\text{SCREW}}$, $N_{\text{edge}}$ is the screw and edge dislocation density and $b$ is the burger vector. Figure 2 shows the AlGaN/GaN HEMT with different Al mole fractions and thicknesses of AlGaN B.B. layer on the MCz-Si

substrate total dislocation results. The 60 nm AlGaN B.B. shows a $5.87 \times 10^9/cm^2$ total dislocation due to the increasing thickness of the AlGaN which was due to the influence of the epitaxy quality of the structure.

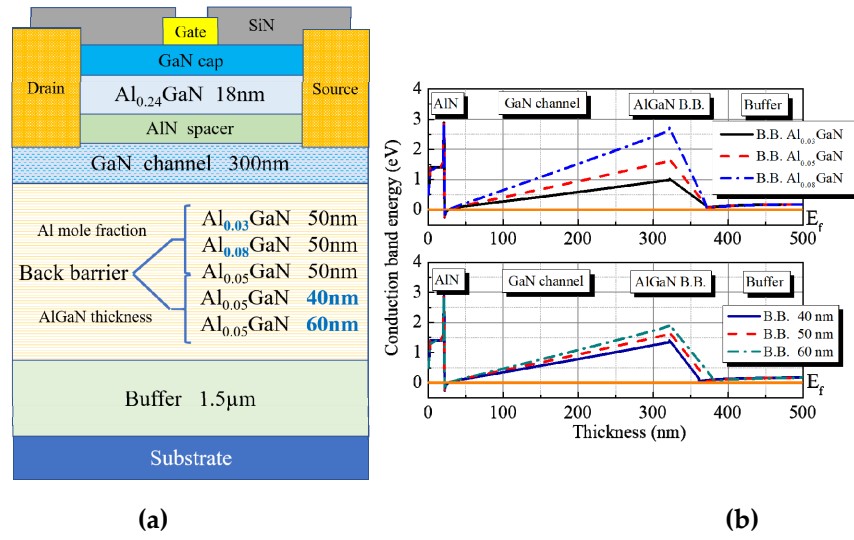

**(a)**                                       **(b)**

**Figure 1.** (**a**) Cross-section of AlGaN/GaN with AlGaN back barrier high electron mobility transistor (HEMT) on Si substrate; (**b**) 1-D poisson simulation.

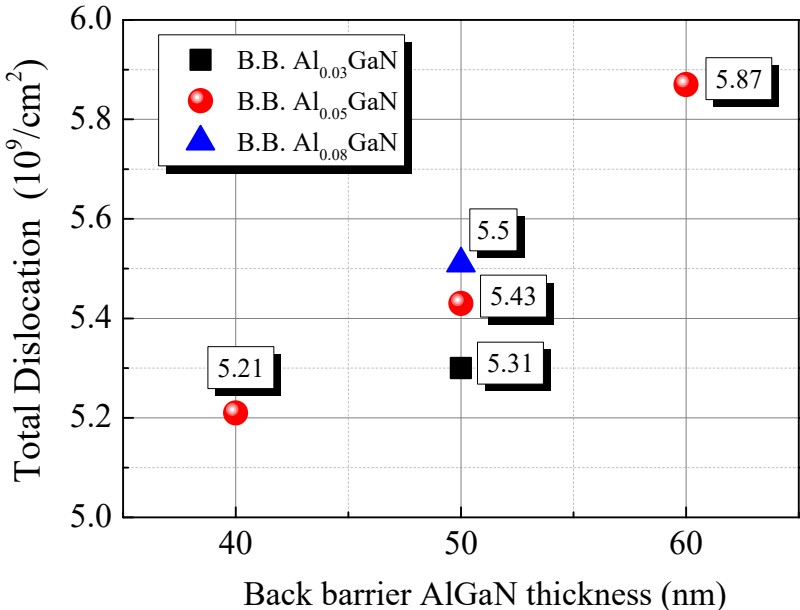

**Figure 2.** The total dislocation of AlGaN/GaN with AlGaN back barrier HEMT on MCz-Si substrate.

## 3. Results and Discussion

Figure 3 shows the characteristics of AlGaN B.B. by varying its thickness and Al mole fraction on the $I_{DS}$-$V_{GS}$ transfer characteristics of the AlGaN/GaN HEMTs with gate voltage ($V_g$) from −6 to 2 V. In Figure 3a and the table, the off-state leakage current was suppressed following the increase of Al mole fraction due to the enhancement of back barrier height. However, the saturation current showed slight degradation with high Al mole fraction B.B. design because the rising of the barrier height also narrowed

down the 2DEG channel. The $I_{on}/I_{off}$ ratios were $6.6 \times 10^4$, $3.2 \times 10^5$, $7.5 \times 10^5$, and the subthreshold swing slopes (S.S.) were 0.28, 0.27, 0.29 V/dec of 3%, 5%, and 8% Al mole fraction devices, respectively. It also affected the transconductance of the device, the $g_{mmax}$ value was decreased from 164 to 156 mS/mm with increasing Al mole fraction. As to the B.B. thickness dependency of device characteristics shown in Figure 3b and the table, the saturation drain current for three samples (40, 50, and 60 nm) were compared. The off-state leakage current of the HEMTs can be improved by increase the B.B. thickness from 40 to 50 nm. It was noticed that the leakage current performances of the device with 60 nm B.B. was worse than 50 nm B.B design and the $g_{mmax}$ value was decreased from 162 to 158 mS/mm with increasing B.B. thickness. As to the results, the epitaxy quality was influenced by thick AlGaN B.B. and the optimized thickness in this study was 50 nm. The devices with 40, 50 and 60 nm AlGaN B.B. showed the $I_{on}/I_{off}$ ratios were approximately $4.5 \times 10^4$, $3.2 \times 10^5$, and $5.2 \times 10^4$ and their corresponding S.S. 0.32, 0.27, and 0.33 V/dec, respectively [19,20].

| | B.B. Al$_{0.03}$GaN | B.B. Al$_{0.08}$GaN | B.B. Al$_{0.05}$GaN 50nm | B.B. 40nm | B.B. 60nm |
|---|---|---|---|---|---|
| S.S. (V/dec) | 0.28 | 0.29 | 0.27 | 0.32 | 0.33 |
| G$_{mmax}$(mS/mm) | 164 | 156 | 160 | 162 | 158 |

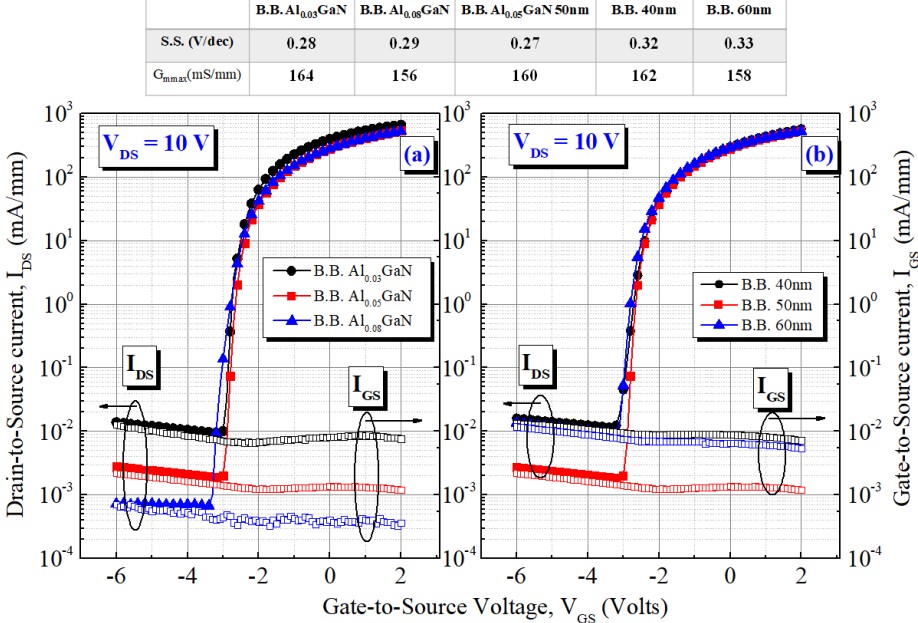

**Figure 3.** The measured $I_{DS}$–$V_{GS}$ at $V_{DS}$ = 10V of devices. (**a**) the different Al mole fraction; (**b**) the different thickness of B.B. devices.

Figure 4 presents the thickness and Al mole fraction variations of the AlGaN B.B. on devices $R_{ON}$ and $I_{DS}$. The measured $V_{GS}$ was 2 V, and the drain voltage ($V_{DS}$) ranged from 0 to 10 V. Based on the results in Figure 4, the table shows that the thickness and Al mole fraction were both negative factors for devices $R_{ON}$ and $I_{DS}$ because AlGaN B.B. played a thermal barrier role between channel and substrate. Thicker B.B. and higher Al mole fraction B.B. thus prevented the junction heat dissipation through the substrate and the devices $R_{ON}$ and $I_{DS}$ were degraded simultaneously.

Figure 5 shows the horizontal breakdown characteristics of the AlGaN B.B. devices. The results were obtained using Agilent B1505. The measurement ohmic contact pattern distance was 40 μm mesa isolation region. Since the increasing Al composition influenced the crystal quality, the breakdown voltage of Al mole fraction 3%, 5%, and 8% AlGaN B.B. device was 740, 729, and 716 V, respectively. The 60 nm AlGaN B.B. device shows that from the highest breakdown voltage it was concluded that B.B. thickness is a positive factor to increase back barrier height and suppress the horizontal leakage current at high

drain bias. The breakdown voltage was improved from 730 to 760 V by increasing the thickness of the back barrier layer.

|  | B.B. Al$_{0.03}$GaN | B.B. Al$_{0.08}$GaN | B.B. Al$_{0.05}$GaN 50nm | B.B. 40nm | B.B. 60nm |
|---|---|---|---|---|---|
| I$_{DSmax}$(mA/mm) @V$_{GS}$=2V | 595 | 542 | 556 | 572 | 548 |
| R$_{on}$(mΩ-mm) | 7.39 | 8.21 | 8.04 | 8.07 | 8.59 |

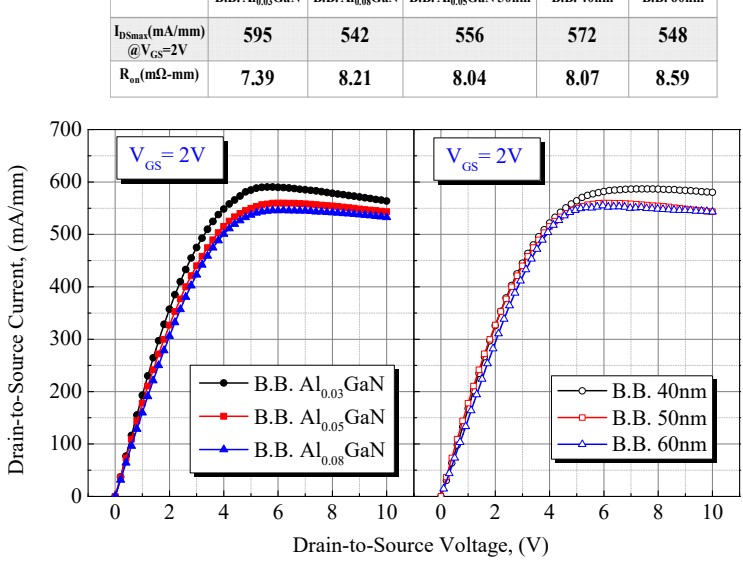

**Figure 4.** The measured I$_{DS}$–V$_{DS}$ at V$_{GS}$ = 2V of devices.

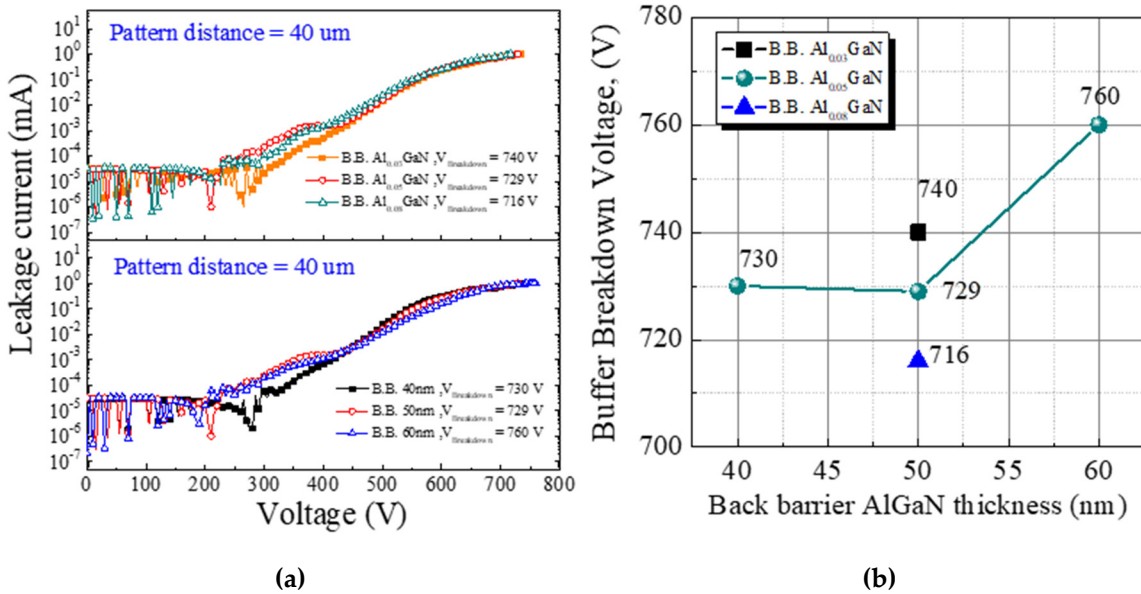

(a)  (b)

**Figure 5.** The horizontal breakdown characteristics of devices. (**a**) The leakage current versus voltage of devices; (**b**) The buffer breakdown voltage of the devices.

To analyze the trapping and detrapping phenomena in GaN HEMTs with AlGaN B.B., low- frequency noise spectra were measured at various gate bias voltages. The drain current power spectral density, $S_{ID}$, was measured and normalized to the square of the drain current versus frequency in the frequency range 1–10,000 Hz. To locate the source of noise in the channel area, the normalized current spectral density $S_I/I^2$ at 100 Hz was plotted against the effective gate-to-source voltage ($V_{GN}$) of AlGaN B.B. HEMTs, where V$_{GN}$ was set as $V_{GS}$–$V_{th}$ [21–23].

When $S_I/I^2$ is proportional to $V_G^{-1}$, there is noise from the heterostructure interface region. In the transition region, where $S_I/I^2$ approaches $V_G^{-3}$, the channel and buffer traps contribute to the noise, the drain-source resistance is dominated by the ungated channel region resistance, $R_U$, which is consistent with a previous study [24–26]. The results (Figure 6) indicated that increasing the Al mole fraction and thickness for the AlGaN B.B. layer induced more lattice dislocation or defects. It results in the $S_I/I^2$ being increased at the $V_{GS}^{-1}$ and $V_{GS}^{-3}$ regions. It is of note that the high Al mole fraction design in AlGaN B.B. layer exhibited the smallest $V_{GS}^{-1}$ region and it was concluded that the channel or buffer traps soon contributed to the noise.

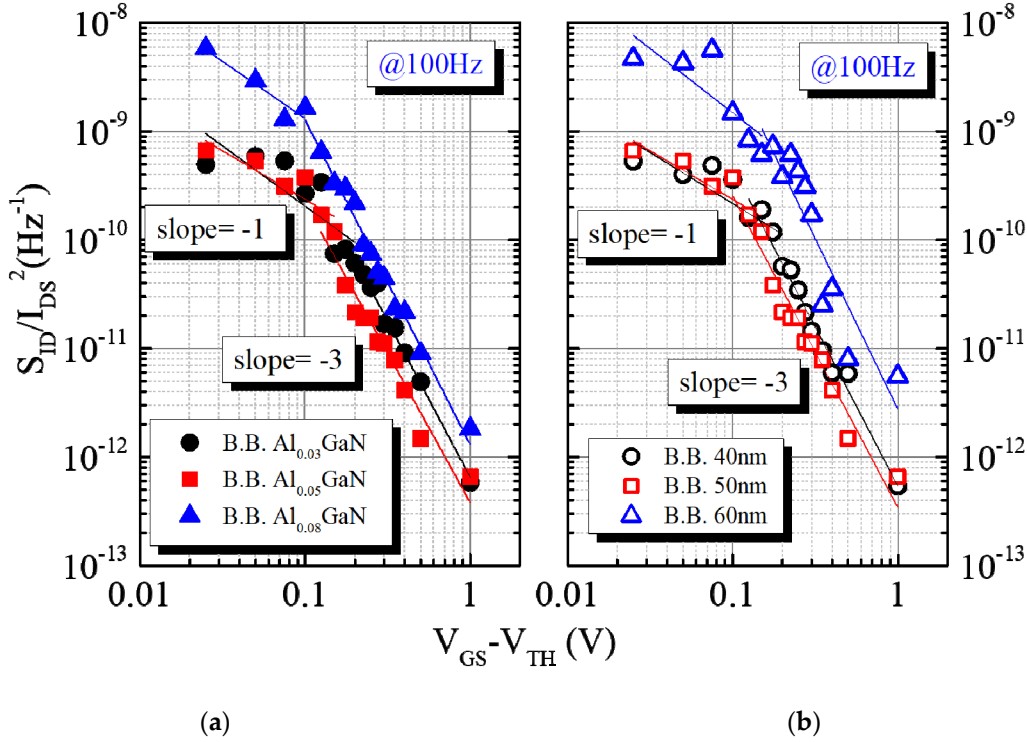

**Figure 6.** The measured of low frequency noise of devices. (**a**) the different Al mole fraction; (**b**) the different thickness of B.B. devices.

The dynamic $R_{ON}$ test was performed using an AMCAD AM241 pulsed system. The dynamic $R_{ON}$ characteristics were measured from different quiescent bias points at $V_{GS}$ = 1 V to investigate the effect of off-state drain bias stress on dynamic $R_{ON}$ and current (Figure 7). The reference bias was set at $(V_{GS,Q}, V_{DS,Q})$ = (0 V, 0 V)—$V_{DSQ}$ is the quiescent drain bias—and did not induce any relevant trapping. The devices were switched with a 5 μs pulse width and 500 μs pulse period. $V_{DS,Q}$ was swept from 0 to 30 V in 10 V increments, and the quiescent gate bias was swept to −6 V.

The Figure 7a shows that the dynamic $R_{ON}$ ratio ($R_{ON,D}/R_{ON,Q}$) of the device with 3% Al mole fraction B.B. under high-field pulse is about 1.26. With Al content increasing, the normalized dynamic $R_{ON}$ of the device with Al$_{0.08}$GaN B.B. increased to 1.31. The increasing number of Al atoms introduce more lattice defects to deteriorate the crystalline quality. This was also affected by the hot electrons in the 2DEG channel which were injected into the buffer traps during the pulse test and caused the on state resistance. The dynamic degradation resulted from insufficient time to release the carriers from the ionized traps [27–30]. Figure 7b shows the ratio also increased when the AlGaN B.B. thickness increased. The thicker AlGaN B.B. devices' dynamic $R_{ON}$ ratio increased from 1.22 to 1.37 at $V_{DS,Q}$ = 30 V.

These results demonstrate that the Al mole fraction and thickness of the B.B. both affected the buffer trapping phenomenon.

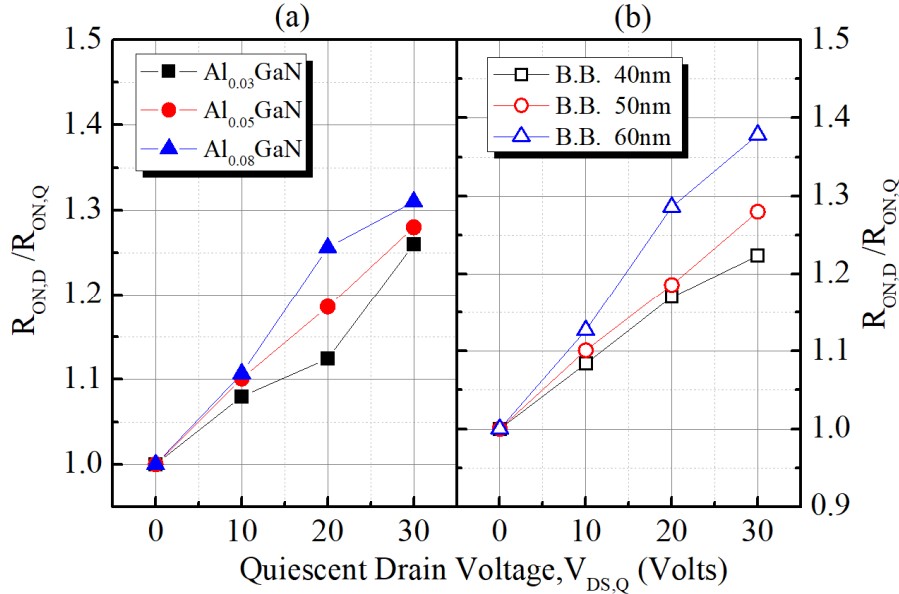

**Figure 7.** The dynamic Ron of devices, (**a**) the different Al mole fraction; (**b**) the different thickness of B.B. devices.

The AlGaN back barrier layer also forms a thermal barrier layer which influences the device linearity and the thermal induced current reduction at high power operation. To experimentally measure the surface temperature distribution in the devices, an infrared (IR) thermographic system with micro-Raman spectroscopy was adopted, and the IR radiation of the device was detected using an IRM P384G detector. The surface temperature map values were obtained from the IR radiation intensity, which was determined following emissivity calibration performed for the B.B. devices at a current of 60 mA for 30 s.

As presented in Figure 8, the surface temperature of the increasing Al mole fraction AlGaN B.B. device was from 48.1 to 50.4 °C. The $Al_{0.08}GaN$ B.B. device had the highest thermal resistance of 30.6 °C/W. The temperature sensitivity is likely related to differences in impurity ionization and carrier hopping between the AlGaN back barrier and buffer structures. Because of increasing the Al mole fraction, the back barrier height also increased as a thermal barrier layer. The results indicated that the Al compositions caused thermal issues. The surface temperature of the thickness of AlGaN B.B. device was from 47.2 to 50.1 °C. Compared to the $Al_{0.08}GaN$ B.B. device, the 60 nm B.B. device had a lower thermal resistance of approximately 29.8 °C/W. The results are related to the device surface temperatures. In addition, the surface temperature distribution was not uniform, which decreased reliability due to the presence of hot spots.

To determine the effect of a B.B. on RF performance, the microwave S-parameters of both devices were evaluated using a common source configuration and a PNA network analyzer in conjunction with Cascade direct probes. The measurement frequency range of the S-parameters was 100 MHz to 50 GHz at the operating condition of $V_{DS}$ = 10 V. The gate length and gate width of devices was 1 and 100 μm. Based on the optimal RF operation of both devices, the $V_{GS}$ values were defined at the appearance of their $g_m$ peak.

As presented in Figure 9, the 8% Al mole fraction of AlGaN B.B. device exhibited a higher current gain cut-off frequency ($f_T$ = 6 GHz). The results indicated that increasing the Al mole fraction can improve

2DEG confinement. This increase affected carrier concentration and the current gain cut-off frequency. However, the maximum stable gain cut-off frequency ($f_{max}$ = 10.5 GHz) was affected by the thermal issues.

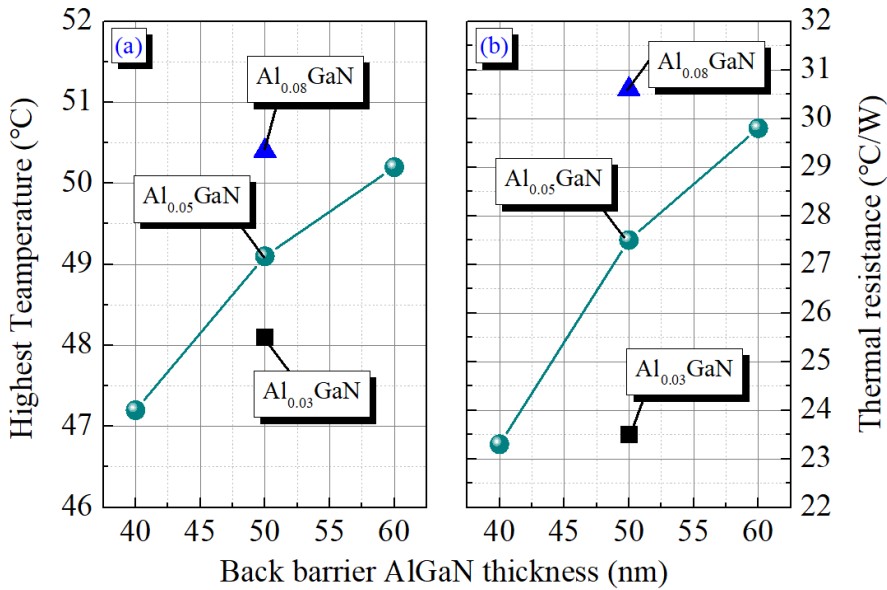

**Figure 8.** (**a**) The peak surface temperature of the device; (**b**) the thermal resistance of the devices.

| | B.B. Al$_{0.03}$GaN | B.B. Al$_{0.08}$GaN | B.B. Al$_{0.05}$GaN 50nm | B.B. 40nm | B.B. 60nm |
|---|---|---|---|---|---|
| F$_T$ (GHz) | 4.9 | 5.2 | 5 | 4.7 | 4.9 |
| F$_{max}$(GHz) | 9.3 | 10.5 | 10.6 | 9.4 | 10.4 |

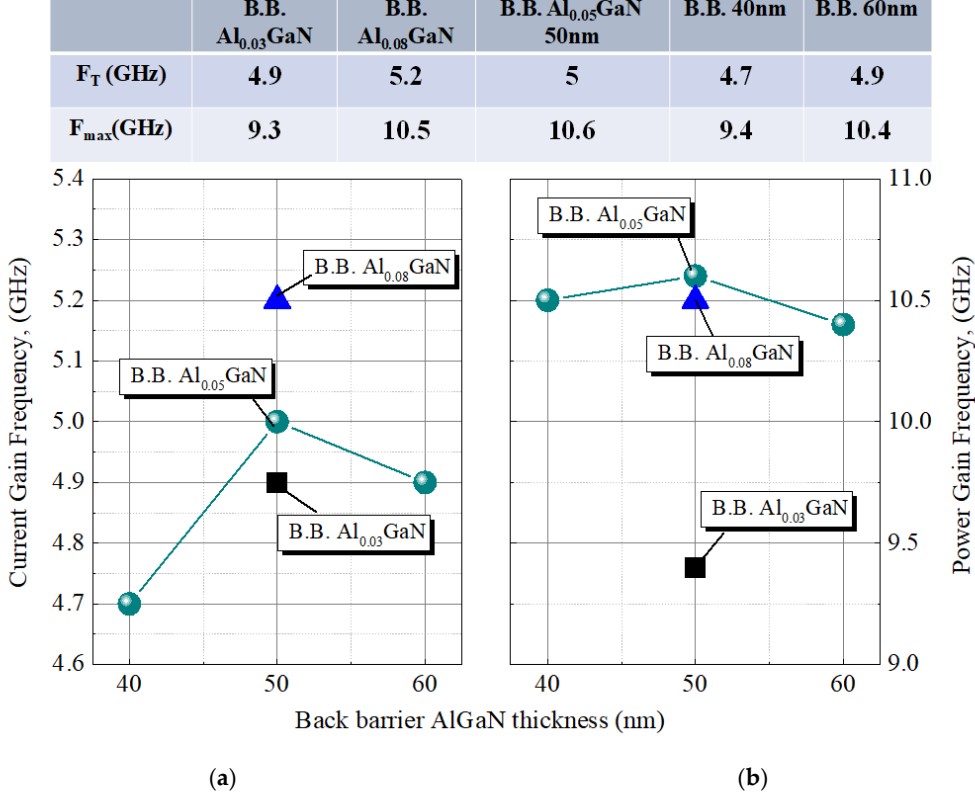

(**a**)  (**b**)

**Figure 9.** The current gain frequency ($f_T$) and power gain frequency ($f_{max}$) of devices. (**a**) the different Al mole fraction; (**b**) the different thickness of B.B. devices.

The thicker AlGaN B.B. influences the epitaxy quality and causes a worse 2DEG confinement. The RF characteristics were not improved with the increasing of thickness of the AlGaN B.B. structure. The 60-nm AlGaN back barrier showed a 4.9 GHz current gain cut-off frequency and 10.4 GHz maximum stable gain cut-off frequency.

## 4. Conclusions

This paper compared various thicknesses and Al mole fractions of AlGaN B.B. in AlGaN/GaN HEMTs. The increasing Al mole fraction of AlGaN B.B. showed that a significant reduction in the $I_{off}$ was obtained because of the prevention of electrons being injected into the buffer. However, the saturation current showed slight degradation at high Al mole fraction B.B. design because the rising of the barrier height also narrowed down the 2DEG channel. The breakdown voltage was improved by increasing the thickness of the AlGaN back barrier. Additionally, the increasing Al compositions of the AlGaN and thickness of the B.B. both affected dynamic $R_{ON}$ and formed a thermal barrier layer which influenced the device thermal induced current reduction at high power operation. The higher Al mole fraction shows a higher current gain cut-off frequency ($f_T$ = 6 GHz) which was improved by 2DEG confinement. However, the maximum stable gain cut-off frequency ($f_{max}$ = 10.5 GHz) was affected by thermal issues. The thicker AlGaN B.B. influences the epitaxy quality causing a worse 2DEG confinement. The RF characteristics were not improved by increasing the thickness of the AlGaN B.B. structure.

**Author Contributions:** Conceptualization, C.Y.C.; Data curation, J.Z.L.; Investigation and writing—original draft preparation, H.Y.W.; Methodology, Y.L.H.; Project administration and writing—review and editing, H.C.C.; Resources, W.C.H.; Validation, C.M.L. All authors have read and agreed to the published version of the manuscript.

**Funding:** This research was funded by the Ministry of Science and Technology (MOST), Taiwan, R.O.C., grant number MOST 108-2218-E-182-006.

**Conflicts of Interest:** The authors declare no conflict of interest.

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
