# Peer review of "The Impact of AlxGa1−xN Back Barrier in AlGaN/GaN High Electron Mobility Transistors (HEMTs) on Six-Inch MCZ Si Substrate"

_coatings, doi:10.3390/coatings10060570_

Round 1

Reviewer 1 Report

The recommedations are enclosed here.

Author Response

The reply letter was attached as a separated letter.

Reviewer 2 Report

This paper shows a study of back barrier in HEMT. Experiements have been duly performed and explained and the results are interesting.

Nevertheless, the English writing must clearly be improved. 

L12-14: “While further increasing the AlGaN thickness could deteriorate the device performance because of the generation of lattice mismatch induced surface defects.”
I do not understand this sentence. What is the subject? What is the verb? If it is related to the previous sentence, it should not be written like that.

L15: “indicated” should be “indicates”.

L16-17: “Thus, the reduction of 16 trapping effect and then the improvement of dynamic RON can thus conclusion.”
I do not understand this sentence.

L18: “influenced” should be “influences”.

L22: “dramatic” … why is it dramatic? Is there a problem with the advancements? Is it bad?

L27: “its low operation”. Since we talk about the CMOS Processes, it should be “their low operation”.

L30: “difficult to be high volume production” … should be “produced in high volume” or something similar.

L31: “…HEMET…” … what is HEMET?

L31: ”…is the adopting…” should be “is the adoption…”

L40: you should say that the MCZ “has been found to be of high interest…”

L47: why do you use “Nevertheless”? I do not understand if the thermal barrier is a good or bad thing for the HEMT. Please give more details, even if you explain that later in the paper.

Fig 1(b) : the mole fractions are difficult to read

L79: “more” should be “mole”.

Fig 2: quality of text font is bad, it is difficult to read

L85: “comparable” should be “compared”

L86: “thickness 40nm to 50nm” should be “thickness from 40nm to 50nm”          

L88: “and its was concluded that the thick AlGaN B.B. influence the epitaxy quality and the optimized thickness in this study was 50 nm.” Should be rewritten.

L90: “were” should be added before “approximately”.

Fig 3: quality of text font is bad, it is difficult to read

Fig 4: too small

L133: “VDS,Q” why is there a comma? And not on L131? Please be consistent.

L139: “it also effected” should be “it was also affected”

L140: “were injected”. The subject is “hot electrons”

L141: “The dynamic …”: what is the subject and the verb in this sentence? It should be rewritten.

L143: “is increasing.”

L144: “affected”, not “effected”.

L160: “are related”

Author Response

(The authors gave the same response as above.)
